# Coffee Leaf Tea from El Salvador: On-Site Production Considering Influences of Processing on Chemical Composition

**DOI:** 10.3390/foods11172553

**Published:** 2022-08-23

**Authors:** Marc C. Steger, Marina Rigling, Patrik Blumenthal, Valerie Segatz, Andrès Quintanilla-Belucci, Julia M. Beisel, Jörg Rieke-Zapp, Steffen Schwarz, Dirk W. Lachenmeier, Yanyan Zhang

**Affiliations:** 1Department of Flavor Chemistry, University of Hohenheim, Fruwirthstr. 12, Verfügungsgebäude 221, 70599 Stuttgart, Germany; 2Coffee Consulate, Hans-Thoma-Strasse 20, 68163 Mannheim, Germany; 3Chemisches und Veterinäruntersuchungsamt (CVUA), Weissenburger Strasse 3, 76187 Karlsruhe, Germany; 4Hochschule für Angewandte Wissenschaften Coburg, Friedrich-Streib-Strasse 2, 96450 Coburg, Germany; 5Finca La Buena Esperanza, Pasaje Senda Florida Norte 124, San Salvador, El Salvador; 6Rubiacea Research and Development GmbH, Hans-Thoma-Strasse 20, 68163 Mannheim, Germany

**Keywords:** coffee leaf tea, novel food, coffee by-products, *Coffea arabica*, caffeine, epigallocatechin gallate

## Abstract

The production of coffee leaf tea (*Coffea arabica*) in El Salvador and the influences of processing steps on non-volatile compounds and volatile aroma-active compounds were investigated. The tea was produced according to the process steps of conventional tea (*Camellia sinensis*) with the available possibilities on the farm. Influencing factors were the leaf type (old, young, yellow, shoots), processing (blending, cutting, rolling, freezing, steaming), drying (sun drying, oven drying, roasting) and fermentation (wild, yeast, *Lactobacillus*). Subsequently, the samples were analysed for the maximum levels of caffeine, chlorogenic acid, and epigallocatechin gallate permitted by the European Commission. The caffeine content ranged between 0.37–1.33 g/100 g dry mass (DM), the chlorogenic acid was between not detectable and 9.35 g/100 g DM and epigallocatechin gallate could not be detected at all. Furthermore, water content, essential oil, ash content, total polyphenols, total catechins, organic acids, and trigonelline were determined. Gas chromatography—mass spectrometry—olfactometry and calculation of the odour activity values (OAVs) were carried out to determine the main aroma-active compounds, which are *β*-ionone (honey-like, OAV 132-927), decanal (citrus-like, floral, OAV 14-301), *α*-ionone (floral, OAV 30-100), (*E*,*Z*)-2,6-nonadienal (cucumber-like, OAV 18-256), 2,4-nonadienal (melon-like, OAV 2-18), octanal (fruity, OAV 7-23), (*E*)-2 nonenal (citrus-like, OAV 1-11), hexanal (grassy, OAV 1-10), and 4-heptenal (green, OAV 1-9). The data obtained in this study may help to adjust process parameters directly to consumer preferences and allow coffee farmers to earn an extra income from this by-product.

## 1. Introduction

Coffee is one of the most consumed beverages in the world. Worldwide, more than 10 million tons are produced each year with a turnover of more than 20 billion US dollars [1]. However, the volatile world coffee market and climate change put pressure on the farmers to keep their farms in good condition. The use of coffee by-products, i.e., products that are created during the production of coffee, could help to create increased income along with increased sustainability [2]. One of these coffee by-products is the coffee leaf, which has been approved as a tea beverage in the EU since July 2020, with maximum permitted quantities of 80 mg/L for caffeine, of 100 mg/L for chlorogenic acid, and of 700 mg/L for epigallocatechin gallate [3]. Coffee leaves are usually produced as a by-product during the pruning of the plants [4].

Coffee leaves from the coffee plant are typically light green (buds and young leaves) to dark green (matured leaves) with a size range of 15 cm (*Coffea arabica*) up to 50 cm (*Coffea liberica*). The lifetime of a leaf is about 7–10 months [5]. Considerable evidence suggests that the leaves of the coffee plant have long been used as a traditional food in the countries where it is grown. Von Pröpper observed in 1882 [6] that “The leaves of the coffee plant, roasted and poured over with hot water, make an excellent tea, which has long been one of the staple foods of the entire Indian archipelago, and is said to be not inferior in effect to the true Chinese tea, but apparently has not yet come into commerce” (authors’ translation from German). Further mentioned in the literature are the countries Ethiopia, West Sumatra, Jamaica, Java, and South Sudan [7,8]. Novita et al. [9] described the traditional production of “Kahwa daun”, a herbal tea from coffee leaves produced in West Sumatra. Herby branches with leaves were clasped on a bamboo stick and then smoked or dried over a cooking fire. In Indonesia, the infusion of coffee leaves is called “copi daon” or “leaf coffee” and in Ethiopia it is called “Quti”; in both countries the leaves are sun dried [10]. Consumption of teas in general may be associated with beneficial health effects, but in most countries, it is consumed just for the taste or the effects of caffeine [11,12,13]. Coffee leaves also contain nutritionally interesting compounds, including carbohydrates, amino acids, organic acids, alkaloids, phenolic compounds, terpenes, carotenoids, phytosterols and flavour compounds such as aldehydes, alcohols, ketones, and esters [14,15,16,17]. The chemical composition of a leaf is highly influenced by the light intensity, nitrogen concentration of the soil, age of the plant and the leaf, growing region and the coffee species [16].

Novel foods or traditional foods from third countries require an authorization to be placed on the market in the European Union (EU) [18]. The application to authorise the placing on the market of an infusion of coffee leaves of the species *Coffea arabica* and/or *Coffea canephora* as a traditional food was approved by the EU Commission on 1 July 2020. Since the applicant could not provide evidence of the use of coffee leaves as an ingredient in other beverages, only the infusion of coffee leaves was approved as a novel food. Critical values have been set for the substances caffeine, chlorogenic acid and epigallocatechin gallate [3].

In this study, the possibility of producing coffee leaf tea in a country of origin were investigated aiming to provide coffee farmers with easily applicable procedures. With the locally available resources, different coffee tea samples were produced. Furthermore, the effects of leaf types, coffee varieties, processing, and drying methods on consumer acceptance, the aroma profile and the product characteristics were studied. This understanding can later help to optimally adjust process and manufacturing parameters to the desired taste. For the analyses of non-volatiles, near-infrared spectroscopy (NIR), high-performance liquid chromatography (HPLC) and nuclear magnetic resonance (NMR) spectroscopy were used. The water content, essential oil, ash content, caffeine, polyphenol content, catechins, organic acids, trigonelline and lactic acid were determined. Afterwards, the samples were sensorically evaluated by a panel according to DIN 10,809 [19], followed by aroma analysis using gas chromatography-olfactometry.

## 2. Materials and Methods

### 2.1. Coffee Leaf Tea Production in El Salvador

The collected leaves are shown in Table 1. The leaf types, their variety, and their collection place are given. Finca La Palma is located in Chinameca, San Miguel and Finca La Quintanilla is located on the north side of Cacahuatique mountain in Morazan, El Salvador. Furthermore, different leaf types are shown in Figure 1.

The harvest of the leaves was conducted between February and March 2021. Old leaves were cut directly from the plant on the field. About five leaves were harvested per plant. All yellow leaves of a plant were picked directly from the branches. Shoots were completely cut off from the plants and then divided into old and young leaves. All leaves were cleaned with fresh water before further processing. For each sample, 600 g of fresh leaves were picked. Typically, about 5–10 leaves were picked per plant, so that about 1000 different plants were picked per variety. For determination of the water content of fresh leaves, one sample of each leaf type was dried in the oven at approx. 80 °C. The weight was determined before drying (approx. 2 g) and after drying with a precision balance. Each experiment was carried out in triplicates.

Different processing steps were applied to obtain a variety of different teas. These steps were related to the processing, drying and fermentation of the leaves. All leaves underwent a withering process where the leaves were stored on drying beds for 12 h overnight at 20 °C. Subsequently, the leaves were either dried whole or processed by various mechanical methods. The methods are shown in Table 2. Furthermore, the processing steps are shown in Figure 2.

#### 2.1.1. Drying Methods

Three different drying methods were carried out for the samples, namely sun/air drying, oven drying and roasting. For sun drying, the leaves were stored on a drying bed for at least 48 h until they were crispy. The oven drying was performed at 70 °C in a gas oven with circulating air for 4 h. The roasting was done on a gas stove until the leaves were crispy. The sun drying process is shown in Figure 3.

#### 2.1.2. Fermentation

For pre-cultivation of the microorganisms (*Saccharomyces cerevisiae* var. bayanus, *Lactobacillus plantarum*), around 1 g of the dry culture was dissolved in 100 mL of water, 1 h before mixing it with the samples. Following that, 40 mL starter cultures were sprayed on the leaves. The leaves were then fermented by storing in closed plastic buckets (anaerobically) for 12 h at a temperature of around 25 °C (overnight). For the wild fermentation, the leaves were stored in the buckets without adding a starter culture.

#### 2.1.3. Postprocessing and Packaging

Samples that had not been already mixed during processing (Table 2) were brought to approximately the same sheet size with the blender. Subsequently, all samples were packed into zip bags and vacuum sealed for transport.

### 2.2. Analysing of Non-Volatile Compounds

#### 2.2.1. Sample Preparation for HPLC, NMR, NIR, and Photometry

The sample preparation of the ground tea sample was carried out according to the international standard ISO 1572 [20]. In accordance with this standard, the samples were prepared using a comminution mill so that the ground material subsequently fell completely through a test sieve with a mesh size of 500 μm. For each sample, a small portion of the sample was first ground in the mill and discarded. Subsequently, the amount of sample required for further testing was ground and packed in a separate, airtight package.

#### 2.2.2. Standard Analytical Procedures

The HPLC analysis of catechins was conducted according to a procedure previously described [21]. For the determination of total phenols, the Folin method was applied using spectrophotometry with a Lambda 35 instrument (PerkinElmer, Rodgau, Germany). For near infrared (NIR) spectroscopic measurement of water, essential oil and ash, a layer of about 1 cm of the tea sample was spread on a Petri dish to cover the entire bottom. Once spread, the sample was pressed firmly with a stamp and placed in the measuring device (Büchi NIRFlex Solids, N-500, Büchi Labortechnik AG, Flawil, Switzerland). Each measurement was performed in triplicate.

#### 2.2.3. Nuclear Magnetic Resonance (NMR) Spectrometry

200 mg of the ground coffee leaf tea was weighed into a 15 mL centrifuge tube and 8 mL of deionised water was added. The tubes were then placed on the combination shaker and shaken on level 8 for 20 min. 2 mL of the solution was membrane filtered into a 4 mL glass vial. Sodium dihydrogen phosphate buffer, pH 6.1, and trimethylsilylpropanoic acid (TSP) were brought to room temperature and 70 μL each of TSP and 100 μL of buffer were pipetted into an empty vial. Then, 600 μL of the membrane-filtered sample was pipetted into each of these prefilled vials. The solution was homogenised before 600 μL of each was pipetted into an NMR tube. The NMR tubes were finally sealed with a lid and a spinner for the NMR instrument. The measurements were performed according to a previously described procedure developed for cold brew coffee [22].

### 2.3. Preliminary Sensory Analysis

Due to COVID-19 pandemic-related contact restrictions during the research period in 2021, only a preliminary sensory analysis was possible using a limited number of tasters. The 24 tea samples from El Salvador of the Bourbon and Pacamara coffee varieties were tasted by a trained test panel composed of 7 people with experience in tea tasting. All respondents have consented to participation in the study. The tasting was done at room temperature with cupping spoons. The main questions of the tasting are: (i) Which teas exhibit the highest popularity among testers? (ii) What flavour profiles do each of the top eight tea samples exhibit?

The coffee leaf tea is prepared according to DIN 10,809 [19] in infusion vessels. After 5 min, the tea is poured off into the bowl and can be tasted after a short cooling period.

The individual samples were rated according to the personal preference of the tasters with values from 0 to 5 (0 = dislikes very badly 5 = likes very well). The 8 highest scoring samples are brewed for a simple descriptive test, a profile test and a rank order test.

The tasted “best eight” were then evaluated in a ranking test by all participants. The samples are ranked from 1 to 8 according to personal preference (1 = best/8 = worst). Multiple assignment of numbers is excluded in this test. A selection must be made even if there are only slight differences (forced-choice test).

The test material was tested for the perceptual attributes of colour, odour, and flavour. Participants were free to add other properties. The individual results were then shared. The terms were collected and either accepted or rejected by the testers. A minimum of 50% agreement was required to define a term.

Finally, the given characteristics (sweet, salty, sour, bitter, body and the dwell time of the taste (finish)) were described with values ranging from 0 (absent) to 5 (strongly expressed).

### 2.4. Analysis of Volatile, Odour-Active Compounds

#### 2.4.1. Tea Preparation

To prepare the tea infusions, 2 g of tea were filled into cellulose bags. These were infused with 200 mL of boiling distilled water in a beaker and infused for 5 min. The tea bag was then removed and the samples were frozen in aluminium bottles at −20 °C.

#### 2.4.2. Direct-Immersion Stir Bar Sorptive Extraction (DI-SBSE)

To extract the compounds from the tea, DI-SBSE technique was applied. Therefore, 10 mL of tea and 3.3 g of NaCl and 0.05 mL of thymol standard (4.2 mg/L) were transferred to a headspace vial (20 mL). The mixture was stirred by a Twister (PDMS) with 1000 rpm for 2 h at room temperature. The Twister was then taken out and rinsed with distilled water and dried off with a lint-free tissue. Afterwards, the Twister was placed in the autosampler of the gas chromatograph. Each Twister was conditioned for 1 h at 250 °C after use. Each measurement was done in triplicate.

#### 2.4.3. Gas Chromatography

Gas chromatography (GC) was performed according to Rigling et al. [23]. In short, an Agilent 7890 B gas chromatograph connected to a 5977 B mass spectrometry detector (Agilent Technologies, Waldbronn, Germany) was equipped with thermal desorption unit (TDU), cooled injection system (CIS) as well as an olfactometry detection port (ODP 3, Gerstel, Mülheim an der Ruhr, Germany). An Agilent J&W DB-WAXms column (30 m × 0.25 mm ID × 0.25 μm film thickness) (Agilent Technologies) was installed. Helium (5.0) served as carrier gas with a constant flow rate of 1.62 mL/min. The gas flow was split 1:1 into the MS detector and the ODP using a μFlowManager Splitter (Gerstel) with a column outlet pressure of 20 kPa. The GC oven temperature was held at 40 °C (3 min), then ramped with 5 °C/min to 240 °C (10 min). The following parameters were applied: MS mode, scan; scan range, *m*/*z* 40–330; electron ionisation energy, 70 eV; source temperature, 230 °C; quadrupole temperature, 150 °C; ODP 3 transfer line temperature, 250 °C; ODP mixing chamber temperature, 150 °C; ODP 3 makeup gas, N_2_ (5.0) (Westfalen). The data were collected using Gerstel ODP1 and Agilent Mass Hunter B07.06 combined with Gerstel Maestro.

Semi quantification was performed using the internal standard thymol (c = 4.2 mg/L) and the weighed-in standard solutions. The response factor of the respective substances could then be calculated using the peak areas of the standard.

#### 2.4.4. Odour Activity Value (OAV)

To determine the odour activity value (OAV) (for details see [23]), the odour threshold of each substance was retrieved from the literature. Values above 1 indicate the possibility of sensory perception of the respective substance.

### 2.5. Statistical Analysis

Microsoft Excel was used to calculate means and standard deviation and for graphical illustrations. The statistical evaluation of the sensory test was performed using the Friedmann test. The calculations (One way ANOVA (confidence level *p* < 0.05) were applied using SPSS (IBM Corporation, Armonk, NY, USA). For the analytical data, the statistical evaluation was carried out using the programme Design Expert 12 (Stat-Ease, Inc., Minneapolis, MN, USA). Hereby an ANOVA for selected factorial model (confidence level *p* < 0.05) was applied. The results are presented as mean value ± standard deviation of the respective parameters.

## 3. Results and Discussion

### 3.1. Moisture Content of Fresh Leaves

The moisture content of the different types of fresh leaves is shown in Table 3. Young leaves and shoots showed the highest water content with 72.89 ± 0.99% and 72.74 ± 1.73%, respectively. Weatherley [24] described a correlation between leaf age and its water content. The water content decreased in all plants during the ageing process. Yellow leaves show the lowest water content with 56.28 ± 1.02%. This effect was to be expected as the plant tries to extract all nutrients and water from the dead leaf before dropping it [25]. To obtain sufficient quantities for the analyses, 600 g of fresh leaves were collected for each sample. Since the pruning of the plants was already done, it was important not to cut off too many leaves and especially not the fresh buds. This could lead to deterioration of the plant [26].

### 3.2. Preparation of Coffee Leaf Tea Samples

In total, 24 different samples were produced during this field study. All samples are shown in Table 4.

### 3.3. Water Content, Essential Oil Content and Ash Content

The results of the NIR analysis for water, essential oil and ash are shown in Table A1 in the Appendix A. Furthermore, the influences of the manufacturing methods are shown in Figure 4.

The water content of the different samples varied from 3.92 g/100 g to 17.42 g/100 g. Statistical analysis of the individual samples showed significant indication of an influence of the leaf type, the processing and the drying method. On average, yellow leaves show the highest water content (8.59 g/100 g), while shoots show the lowest water content (5.83 g/100 g). In processing, freezing (6.58 g/100 g), the use of the whole leaf (6.65 g/100 g) and blending (7.93 g/100 g) lead to lower water content. These values are up to 33% lower compared to the other processing steps. The different surface area plays a role here. Blended samples will dry much faster than less processed ones [27]. Furthermore, up to 46% lower water contents could be achieved during drying by roasting and by the oven. According to Arslan et al. [28], oven drying has a more than double drying rate compared to sun drying. However, the water content of sample 16 with the highest water content was significantly higher compared to the other samples, indicating an error during sun drying. Here, the sample was probably removed from the drying bed too early. Furthermore according to German guidelines [29], the water content of a tea or tea-like product must not exceed 8%. In future production, special care must be taken to ensure that the sun-drying process is not terminated too early.

The essential oil content of the leaves in this experiment varies from not detectable to 1.52 mL/100 g. The statistical analysis showed a significant influence of the variety and the drying method. Leaves from Pacamara have a significantly higher oil content compared to Bourbon. The effects of drying methods on the essential oil content have already been investigated in many previous studies for different plants. Here, gentle drying in the oven resulted in a higher oil content than drying under direct sunlight [30,31,32].

The ash content of the leaves showed no significant influencing factors. The values of all samples ranged between 7.81 g/100 g and 10.20 g/100 g regardless of the influences.

### 3.4. Content of Caffeine and Catechins

The results of the HPLC analysis for caffeine and catechins are shown in Table A2 in the Appendix A. Furthermore, the influence of the manufacturing methods on caffeine and the total catechins are shown in Figure 5 and Figure 6.

#### 3.4.1. Caffeine

The caffeine content of the leaves varies between 0.37 g/100 g DM and 1.33 g/100 g DM. Here, the leaf type and the variety show significant influences on the caffeine content of the tea. Young leaves show the highest caffeine value (0.91 g/100 g DM) while yellow leaves show the lowest (0.44 g/100 g DM). The caffeine levels are approximately the same as those detected by Ratanamarno et al. [33]. The effect of caffeine reduction with leaf age has already been observed in some studies with different plants [34,35]. Song et al. [36] explained this effect mainly by the function of caffeine as a pesticide. Younger leaves of the plant must be more protected compared to old ones; therefore, the plant builds up higher concentrations in those leaves. Furthermore, the processing is a significant variable towards the caffeine content. Here, rolling (1.20 g/100 g DM) and freezing (1.00 g/100 g DM) of the leaf show the highest contents of caffeine. According to Astill et al. [37], the caffeine content decreases during the fermentation and drying stage. In case of freezing, it is possible that metabolic pathways are stopped, which result in less degradation during drying. Since the rolling process was only carried out on young leaves, further tests would be required to determine whether this influences the caffeine content. Furthermore, the low caffeine content of the blended samples may be due to the addition of water. Some of the caffeine may have been dissolved in the water during processing and then dripped off through the drying bed during drying. Theobromine, which is described in the metabolic pathway as a precursor of caffeine [38], could only be detected in small amounts in three samples.

#### 3.4.2. Catechins

The results of the total catechin content show a clear influence of the blending process. All blended samples, with the exclusion of sample 6, have no catechins at all. According to scientific findings, this could be due to the oxidation of catechins by polyphenol oxidase to theaflavin [39]. This effect occurs during the fermentation of black tea [40]. The large surface area and added water of the mixed samples could be responsible for an enhanced enzymatic reaction. Furthermore, the drying parameters show an influence on the total catechin content. Air drying has a significantly higher average value (0.266 g/100 g DM) than oven drying (0.054 g/100 g DM) and a significantly lower value than roasting (0.479 g/100 g DM). Li et al. [41] investigated the correlation of temperature and duration of thermal processing on catechin content. Accordingly, the low content of the oven-dried sample can be attributed to the 4 h drying time. The roasted sample had a much shorter drying time (20 min), which resulted in the highest content.

The epigallocatechin gallate mentioned in the European Commission’s novel food approval could not be detected in any sample. Since Ratanamarno et al. [33] have already detected epigallocatechin gallate in fresh coffee leaves, it can be concluded that it was degraded during processing, transport or storage. Turkmen et al. [42] observed an absence of epigallocatechin gallate in black tea, attributed to oxidation and fermentation processes [42]. This effect could also have occurred in the coffee leaf tea samples during the withering process.

### 3.5. Content of Total Polyphenols

The results of the photometric analysis of the total polyphenol content are shown in Table A3 in the Appendix A. Furthermore, the influence of the manufacturing methods is shown in Figure 7.

The total polyphenol content varies between not detectable and 10.36 g/100 g DM and depends mainly on the processing. The blended samples show a significantly lower phenol content with 2.55 g/100 g DM compared to crumbling, cutting, rolling and whole leaves. Furthermore, no phenol could be detected in the frozen sample. The existing literature shows the influence of freezing on the polyphenol content. Oszmiański et al. [43] found a loss of up to 33.6% and Loncaric et al. [44] up to 48% after a freezing process. The low content in the blended samples can be explained by an increased oxidation process. Due to the large surface area of the blended leaves, the polyphenol oxidase can degrade the polyphenols faster compared to the other samples. Turkmen et al. [42] described a decrease in polyphenol content in fermented black tea compared to green tea. Here, the polyphenol oxidase has not been deactivated by a heat process, resulting in a loss that depends on the duration of the fermentation.

### 3.6. Content of Organic Acids and Trigonellin

The results of the NMR analysis for organic acids and trigonellin are shown in Table A4 in the Appendix A. Furthermore, the influence of the manufacturing methods is shown in Figure 8, Figure 9, Figure 10 and Figure 11.

#### 3.6.1. Chlorogenic Acid

The content of chlorogenic acid (3-caffeoylquinic acid) varies between not detectable and 9.35 g/100 g DM. The significant influences here are the leaf type and the processing. The young leaves show the highest amount of chlorogenic acid with 5.33 g/100 g DM followed by shoots with 1.96 g/100 g DM, the old leaves with 1.71 g/100 g DM, and the yellow leaves with 1.21 g/100 g DM. As with caffeine, chlorogenic acid exerts a protective effect on the leaf through its antioxidant property. Therefore, here it is also present in an increased amount in the young leaves. These data also coincide with the analyses already carried out by Monteiro et al. [45]. Furthermore, it is shown that freezing (0.31 g/100 g DM) and blending (0.47 g/100 g DM) have a negative effect on the amount. The negative effect of freezing contradicts a study by Ścibisz et al. [46] where freezing had no effect.

#### 3.6.2. Lactic Acid

Lactic acid varied from 0.11 g/100 g DM to 8.12 g/100 g DM in the samples. As expected, the samples fermented with the *Lactobacillus* showed the highest value. Furthermore, an increased amount is found in the wild fermented samples. This suggests that a certain percentage of lactic acid bacteria is present in the microbiota of the coffee leaf.

#### 3.6.3. Acetic Acid

Acetic acid varied from 0.04 g/100 g DM to 0.29 g/100 g DM. Above all, the processing and the type of leaf influence the amount significantly. As with the other acids, freezing gives a significantly lower value (0.04 g/100 g DM) than the rest. Samples with larger surface area such as blended and cut samples show the highest values (0.14 g/100 g DM and 0.13 g/100 g DM). Samples fermented wild and with *Lactobacillus* have significantly higher values (0.17 g/100 g DM/0.18 g/100 g DM) than those fermented with yeast or not fermented (0.09 g/100 g DM). The yeast fermented samples do not differ from the non-fermented ones, indicating that this yeast strain produces few organic acids.

#### 3.6.4. Trigonelline

Trigonelline is a substance, which is mainly found in the seeds of many plants. Quantities between 0.62 g/100 g DM and 4.87 g/100 g DM could be detected in the tea samples. The trigonelline content results in this study are up to four times higher than those found in a recent study by Monteiro et al. [45]. In the coffee beans, however, only 1–1.2 g/100 g DM are contained in the untreated state. The trigonelline content is influenced by the leaf type and the drying method. The higher content in young leaves coincides with the result of Monteiro et al. [45]. Furthermore, the roasting leads to a high trigonelline content. Zhu et al. [47] showed a similar relationship with hemp seeds where roasting had the highest influence on trigonelline levels.

### 3.7. Preliminary Sensory Evaluation

The following results of the sensory evaluation must be considered as being preliminary due to a restricted taste panel (*n* = 7).

#### 3.7.1. Personal Acceptance

The results of the ranking test with the eight best rated teas are shown in Table 5. An evaluation of the results using the Friedmann test showed no statistical differences between the teas. Nevertheless, an improvement in flavour by yeast fermentation can be inferred by these results. The top 2 were both treated with the Anaferm yeast. For sample 369, however, a strong polarization could be detected within the panel. It was described as “the best” by two people and as “the most disliked” by two other people from the panel.

#### 3.7.2. Simple Descriptive Test

The results of the simple descriptive test are shown in Table 5. They show a wide range of flavours perceived by the panel. It can be recognised how the fermented samples 182, 687, 156 and 147 differ from the unfermented samples in the type of aromas. While fermentation tends to produce sweetish fruity notes, the unfermented samples tend to have green and vegetal aromas. Wang et al. [48] also found a correlation between fermentation and a loss of green flavour compounds and an increase of fruity flavours. There was also a wide difference in the various colours. Here, as shown in Figure 12, the fermented teas especially show a considerably darker colour compared to the rest. The exception is sample 147, which despite fermentation has the lightest colour of all the teas. The change of colour was also reported by Borah et al. [49]. In the study, the tea changed the colour from green to a darker copper red during the fermentation process.

#### 3.7.3. Profile Test

The results of the profile test (Figure 13) show that the taste is mainly dominated by sweet and partly bitter impressions. Sweetness ranged from 2.8 to 4 in all samples, bitterness ranged from 1.5 to 2. Acidity was only detected with a value of 0.8 in sample 182 and saltiness could not be detected. Values of less than 0.5 were not considered in this study. This coincides with the description of Yuwono et al. [50], in which the authors describe the tea as sweetish, green and woody. The evaluation with the software programme Design Expert showed no significant influence of the different process parameters on sweetness, saltiness, bitterness, acidity, body and finish.

### 3.8. Aroma Analysis via DI-SBSE-GC-MS-O

#### 3.8.1. Identification of Odour-Active Compounds

A total of 68 different olfactory impressions could be detected from the 8 samples (Table A5). Of these, 44 could be identified by mass spectrum, RI, and odour. Exemplary total ion chromatograms and mass spectra are shown in Figure A1 and Figure A2. Of the 44 substances identified, 16 are aldehydes, 10 are ketones, 8 are alcohols, 3 are organic acids, 2 are pyrazines, 2 are ionenes, 1 is a terpene, 1 is an aromatic heterocyclic amine, and 1 is a fatty acid ester. A total of four substances were detected in all samples. These are (*E*,*E*)-3,5-octadien-2-one, 2,6-nonadienal, *α*-ionone and *β*-ionone. Pyrazines, which are typical roast aromas, could only be identified in the roasted sample 369. 4-Heptanal could only be detected in fermented samples and *γ*-dodecalactone just in yeast fermented samples.

The perceived odours could be described to a large extent as green and grassy (22 substances). Furthermore, some sweetish notes (11 substances) and notes in the area of melon or cucumber (11 substances) could be identified. In the area of fruity and citrus, 7 and 5 impressions, respectively, were perceived. Other attributes were roasty, herbal, honey, vanilla, aquarium, unique, nutty, forest and stable.

#### 3.8.2. Semi-Quantification and Odour Activity Values

The results of the semi-quantification and the calculated OAVs of each sample are shown in the Table A6, Table A7, Table A8, Table A9, Table A10, Table A11, Table A12 and Table A13 in the Appendix A. The associated OAVs can be found in Table A14 in the Appendix A. Additionally, the percentage of the OAVs in the total aroma for each sample is shown in Figure 14, Figure 15, Figure 16, Figure 17, Figure 18, Figure 19, Figure 20 and Figure 21.

Sample 182 showed a total of nine substances with an OAV greater than 1. Three of these substances, *β*-ionone (54.5%), (*E*,*Z*)-2,6-nonadienal (28.4%), and *α*-ionone (11.1%) accounting for 94.0% of the total aroma. Furthermore, the substances decanal, 4-heptenal, hexanal and (*E*)-2-nonenal are partly responsible for the aroma.

The smell of the tea was described in the previous tasting as floral and woody while the flavour was sweetish and peach via retronasal detection. The sweetness here can most likely be attributed to the *β*-ionone while the green and floral tones come from (*E*,*Z*)-2,6-nonadienal and *α*-ionone. Another role is probably played by hexanal (0.7%) which, according to Zhu and Xiao [51], is one of the key aroma compounds of peach. These, together with decanal (fruity) and octanal (citrus like) could have caused the perceived peach-like flavour.

Sample 687 shows six compounds with an OAV over 1. Here the main compounds are *β*-ionone (40.3%) and (*E*,*Z*)-2,6-nonadienal (28.7%). Furthermore, decanal (8.2%), octanal (7.7%) (*E*,*Z*)-2,4-nonadienal (7.5%) and *α*-ionone (7.5%) have a relatively large share. Overall, however, the sample showed significantly lower total concentrations than all other samples. This could be because the plant has extracted the substances from the yellow leaves before dropping them [52]. The resulting high percentage of fruity substances, such as (*E*,*Z*)-2,6-nonadienal and (*E*,*Z*)-2,4-nonadienal most likely leads to the detected fruity peach-like flavour of the tea.

A total of 9 aroma compounds with an OAV more than 1 were detected in sample 234. The main aroma substances are *β*-ionone (64.7%), decanal (13.9%) and *α*-ionone (12%). Furthermore, the substances (*E*,*Z*)-2,6-nonadienal, octanal, 2,4-nonadienal, 4–heptenal, (*E*)-2-nonenal and hexanal are partly responsible for the aroma. The strongly pronounced floral tones can therefore probably be attributed to decanal and *α*-ionone, together with (*E*,*Z*)-2,6-nonadienal (4.7%), hexanal (0.5%) and 4-heptanal (0.6%). The sensory tones of the green bean and the rooibos do not coincide here, however, with the aroma components of these two products described in the literature [53,54].

In sample 156, 9 aroma compounds could be identified with an OAV of more than 1. *β*-ionone (39.2%) and decanal (34.4%) form the main part of the aroma compounds. Furthermore, (*E*,*Z*)-2,6-nonadienal with 10% and *α*-ionone with 9.1% are main aroma compounds, while the substances octanal, 2,4-nonadienal, (*E*)-2-nonenal, hexanal and 4-heptenal account for the remaining 7.3%. Overall, the total concentrations of the various compounds are significantly higher compared to the comparable sample 687, which also consists of yellow leaves and was fermented. The difference here is in the type of fermentation, which in this sample was wild and uncontrolled. Microorganisms could have caused the increase in concentrations in this sample; however, this would have to be verified by further studies.

Similar to the previous sample, 9 aroma forming substances were detected in sample 930. Here, *β*-ionone (49.2%) and decanal (35.1%) make up the main component. Including *α*-ionone (5.4%), the substances (*E*,*Z*)-2,6-nonadienal, 2,4-nonadienal, octanal, (*E*)-2-nonenal, hexanal and 4-heptenal account for a total of 15.6% of the aroma. The aroma described as sweetish and floral is assumed to be composed mainly of the substances *β*-ionone, decanal, *α*-ionone, hexanal and 4-heptenal. Furthermore, no influence of the heat process (steaming) on the volatile substances can be identified for this sample.

Semi-quantification also identified 9 compounds of importance for the aroma in sample 743. *β*-Ionone and *α*-ionone are the main substances here with 75.1% and 9.1%. Furthermore, decanal (6.5%), (*E*,*Z*)-2,6-nonadienal (5.4%) and with 1% or less 2,4-nonadienal, octanal, hexanal, 4-heptenal,(*E*)-2-nonenal are odour-active. The aroma described as grassy and floral is assumed to be composed mainly of the substances, decanal, *β*-ionone, *α*-ionone, hexanal and 4-heptenal.

A total of 8 different substances with an OAV higher than 1 were detected in sample 147. The main flavouring agent is *β*-ionone with 80.8%. This is followed by *α*-ionone (7.7%), decanal (6.4%) and the others, (*E*,*Z*)-2,6-nonadienal, octanal, hexanal, 2,4-nonadienal, and 4-heptenal with 5.1%. In general, the sample had the highest concentration of *β*-ionone (6.49 µg/L, Table A12). This high level of *β*-ionone also seems to be confirmed in the odour perception, which would be described as chestnut blossom, and the flavour, which was determined by the panel to be honey.

In roasted sample 369, a total of 9 aroma-active compounds with an OAV above 1 could be detected. Here, *β*-ionone is the main aroma substance with 86.5%. Furthermore, 2,3-dimethylpyrazines (0.3%) and 2-ethyl-3,5-dimethylpyrazines (0.6%) with an OAV of 1 and 2 are found in the sample. However, the smoky and roasted flavour of this tea is strongly influenced by these components despite their minute contribution to the overall aroma. One reason for the strong perception of smoke is a natural protective instinct of humans against fire. The sensory cells perceive it very strongly in order to detect the danger of fire at an early stage [55].

In total, 9 main aroma compounds (Figure 22) were identified in coffee leaf tea via semi quantification and calculation of OAV.

Three of them, *β*-ionone, *α*-ionone and 2-ethyl-3,5-dimethylpyrazines are, according to Ho et al. [57], also main aroma substances of the tea plant. *β*-Ionone has the largest overall share in all samples and varies between 39.2% (156) and 86.5% (369). Furthermore, the percentage of decanal in two samples with a *β*-ionone value below 50% is very high, i.e., 34.4% (156) and 35.1% (930). *α*-Ionone which is also present in all samples varies from 5.4% (930) to 12% (234). Moreover, (*E*,*Z*)-2,6-nonadienal was found as a main aroma component in some samples. As in the case of decanal, the value here varies greatly between 1.8% (147) and 28.7% (687) depending on the sample. The citrusy note of (*E*)-2-nonenal could not be noted in any of the samples during tasting by the panel. Despite the large variation between the samples, it was not possible to correlate the differences with the parameters in this study. Further studies are required to provide comparable conditions between samples. Only the roasting process showed a large difference in these tests compared to the other samples, as the pyrazines also contribute a large proportion to the aroma.

The main disadvantage of the calculated OAV is that the interaction of individual aroma components is not considered. It is possible that individual aroma components, which were excluded because of an OAV below 1, may well contribute to the aroma together with other substances [58]. Additionally, the perceived intensity of an aroma is not proportional to the value of the OAV. A doubling of the OAV around the threshold can have a greater impact on the aroma than a doubling of an OAV very far above the threshold [59].

## 4. Conclusions

In this study, different ways of producing teas from coffee leaves directly on the farm were investigated. The samples differed both in popularity among consumers and in the chemical composition of the active ingredients and flavourings. In future, the data obtained in this study may help to adjust process parameters directly to consumer preferences and allow farmers to earn an extra income from this by-product. For this purpose, on-site experiments should be carried out to upscale the processes.

Young leaves showed a positive correlation on various plant protective ingredients such as caffeine content, chlorogenic acid and trigonelline. The variety played a role in essential oils and caffeine content in these experiments. Pacamara had an increased level of essential oil and a slightly lower level of caffeine compared to Bourbon. In the processing parameters, blending of the samples resulted in a strong decrease in caffeine, catechins, polyphenols and chlorogenic acid. In contrast, cell disruption processes such as rolling or crumbling led to increased values. Among the drying methods, roasting and oven drying had a positive effect on the essential oil and trigonelline content in the samples. Fermentation mainly affected the acidity of the samples. Increased levels of lactic and acetic acid were found here, particularly in wild fermented and *Lactobacillus* fermented samples.

The sensory analysis revealed that fermented teas in particular are ahead in terms of popularity with consumers. Green tones in particular are masked by more fruity notes (e.g., peach). Additionally, the roasting of the tea seems to lead to a polarizing product.

In the aroma analysis by gas chromatography, 68 aroma active compounds were detected by the ODP. By calculation of the OAV, 6–9 aroma compounds could be determined for each tea, which are the main components of the aroma profile. These are *β*-ionone (honey-like), decanal (citrus, floral), *α*-ionone (floral), (*E*,*Z*)-2,6-nonadienal (cucumber-like), 2,4-nonadienal (melon-like), octanal (fruity), (*E*)-2 nonenal (citrus), hexanal (grassy) and 4-heptenal (green). Additionally, the two substances 2,3-dimethylpyrazines and 2-ethyl-3,5-dimethylpyrazines were found in the roasted sample.

## Figures and Tables

**Figure 1 foods-11-02553-f001:**
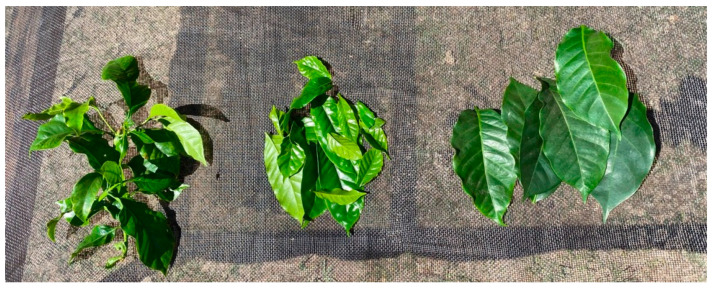
From left to right: whole shoots, young leaves, old leaves (Bourbon Tekisic).

**Figure 2 foods-11-02553-f002:**
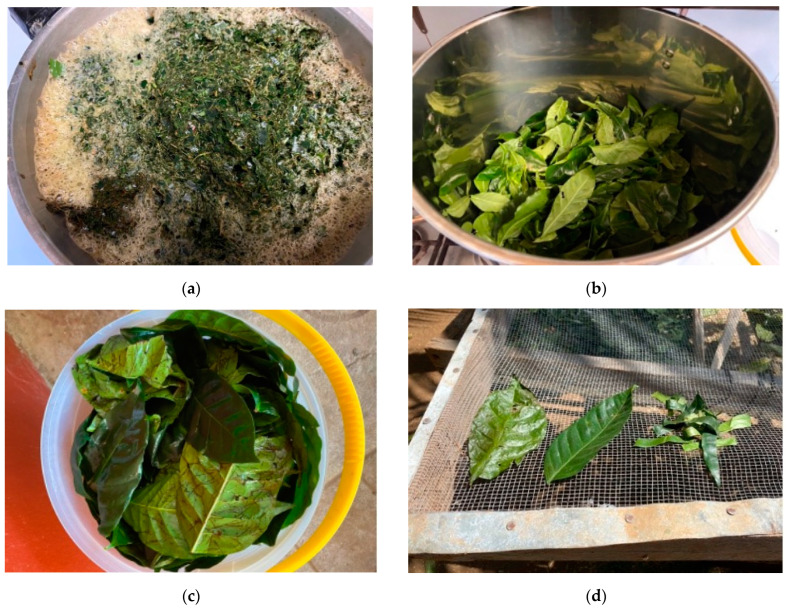
(**a**) Leaves after blending. (**b**) Leaves in the steaming pot. (**c**) Leaves directly after crumbling. (**d**) Crumbled, whole and cut leaves on the drying bed.

**Figure 3 foods-11-02553-f003:**
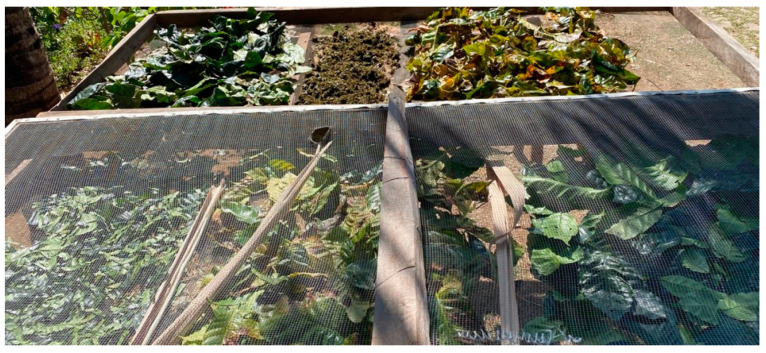
Sun drying of the different samples on the drying bed in Chinameca.

**Figure 4 foods-11-02553-f004:**
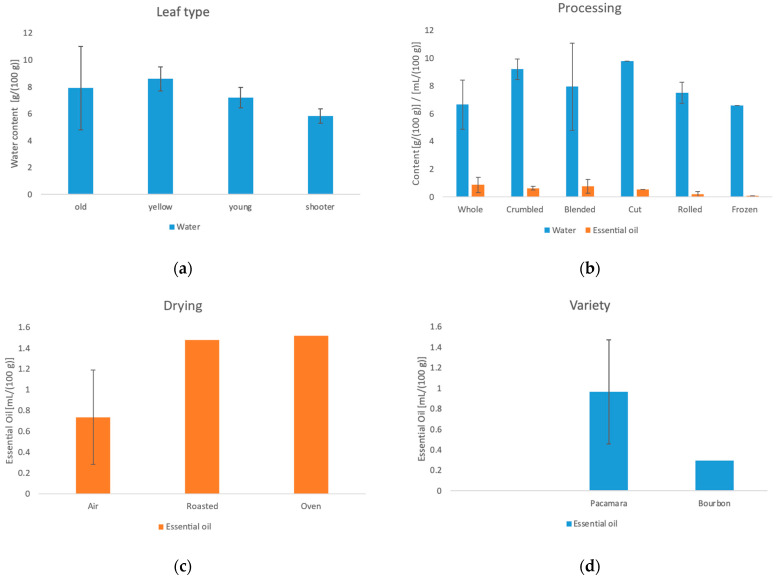
(**a**) Influence of the leaf type on the water content. (**b**) Influence of the processing methods on the water (g/100 g) and the essential oil content (mL/100 g). (**c**) Influence of the drying method on the essential oil content. (**d**) Influence of the variety on the essential oil content.

**Figure 5 foods-11-02553-f005:**
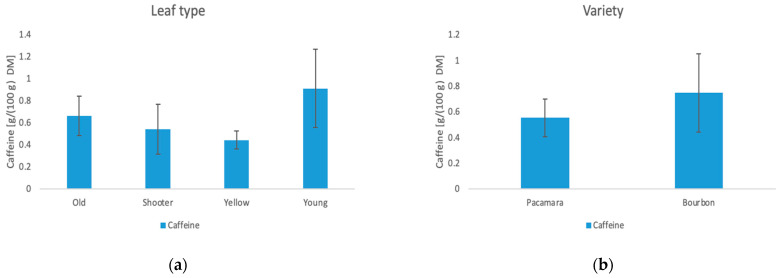
Influence of the leaf type (**a**), the variety (**b**), and the processing method (**c**) on the caffeine content.

**Figure 6 foods-11-02553-f006:**
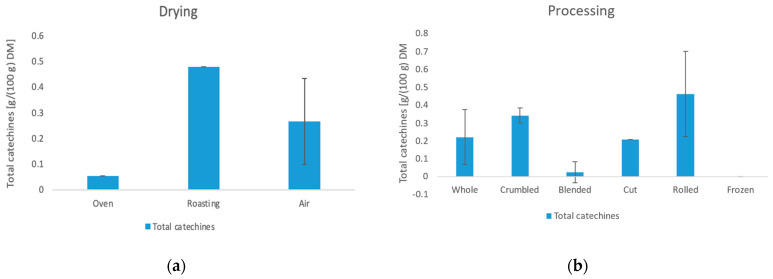
Influence of the drying method (**a**) and the processing method (**b**) on the total catechin content.

**Figure 7 foods-11-02553-f007:**
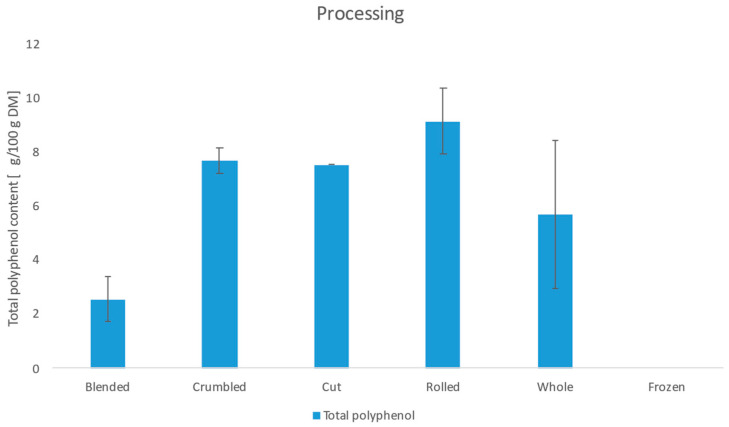
Total polyphenol content depending on process parameters.

**Figure 8 foods-11-02553-f008:**
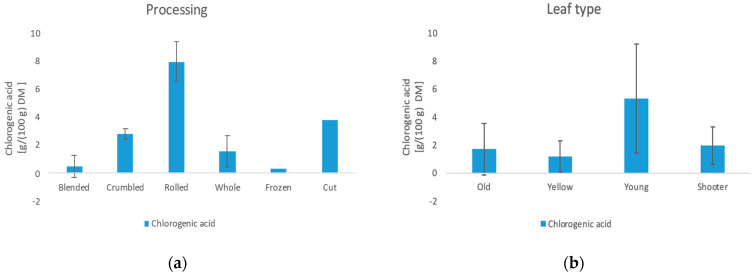
(**a**) Influence of the processing method on the chlorogenic acid content. (**b**) Influence of the leaf type on the chlorogenic acid content.

**Figure 9 foods-11-02553-f009:**
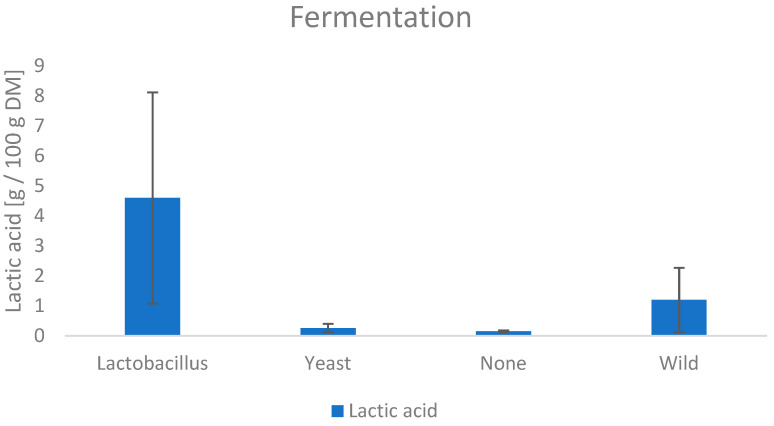
Influence of the fermentation on the lactic acid content.

**Figure 10 foods-11-02553-f010:**
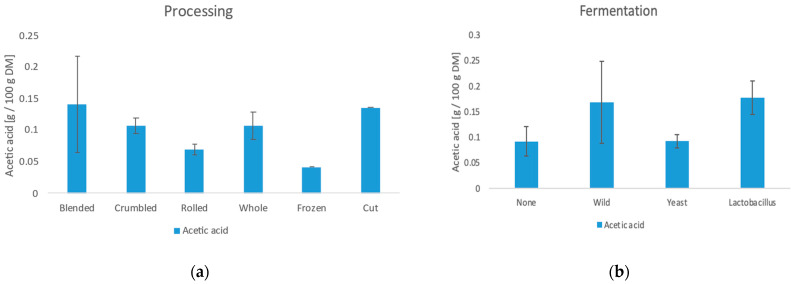
(**a**) Influence of the processing method on the acetic acid content. (**b**) Influence of the fermentation on the acetic acid content.

**Figure 11 foods-11-02553-f011:**
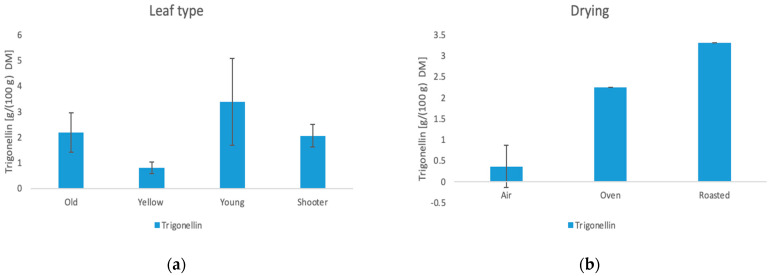
(**a**) Influence of the leaf type on the trigonelline content. (**b**) Influence of the drying method on the trigonelline content.

**Figure 12 foods-11-02553-f012:**
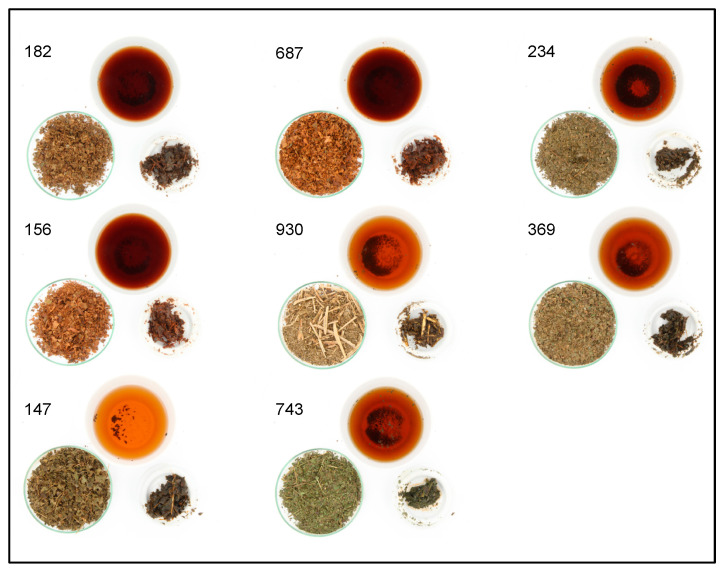
Brewed tea samples along with brewed and unbrewed tea leaves of the best eight teas.

**Figure 13 foods-11-02553-f013:**
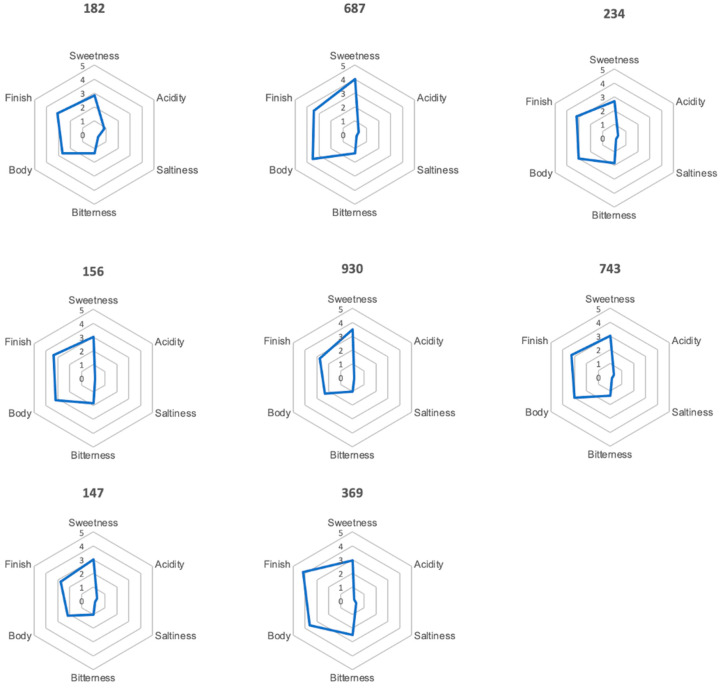
Graphical illustration of the results of the profile test.

**Figure 14 foods-11-02553-f014:**
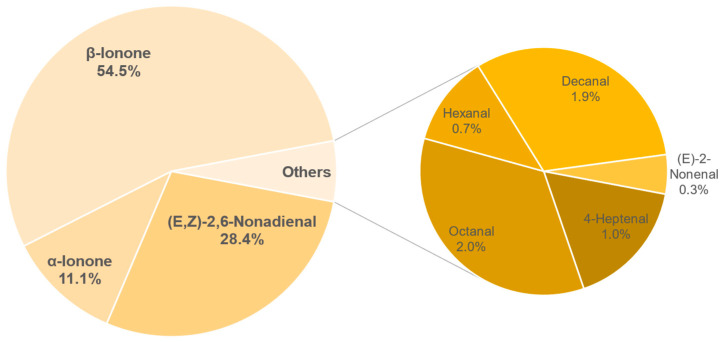
Percentage of OAV of each compound in the total aroma of sample 182.

**Figure 15 foods-11-02553-f015:**
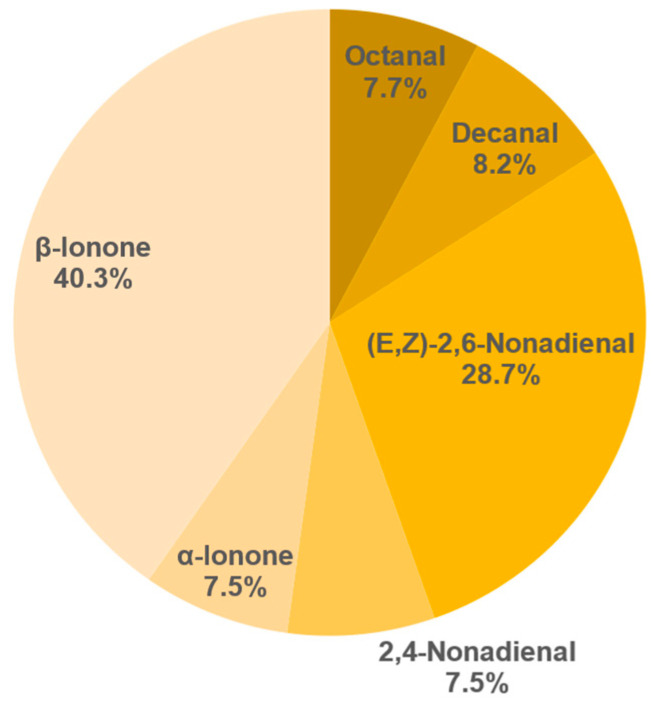
Percentage of OAV of each compound in the total aroma of sample 687.

**Figure 16 foods-11-02553-f016:**
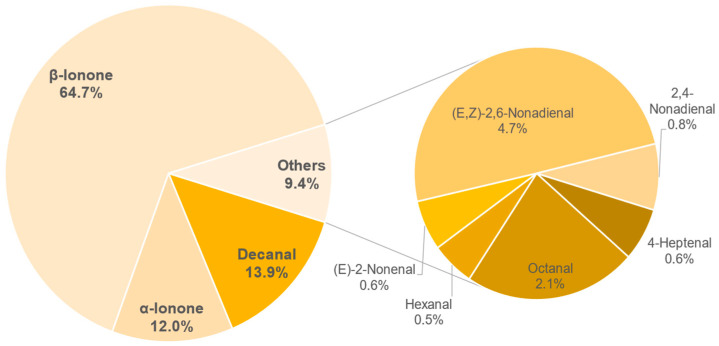
Percentage of OAV of each compound in the total aroma of sample 234.

**Figure 17 foods-11-02553-f017:**
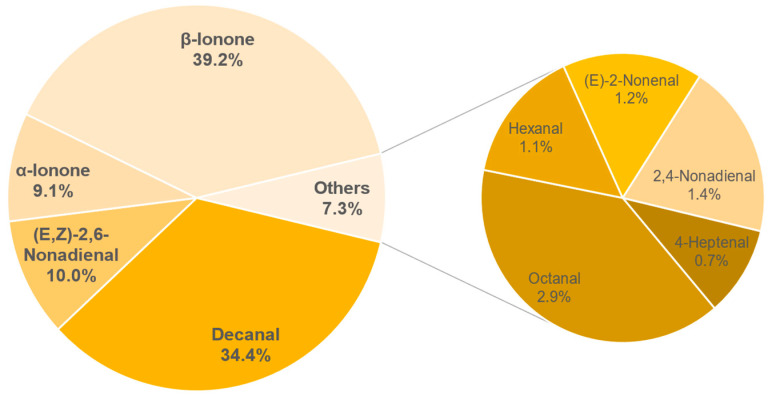
Percentage of OAV of each compound in the total aroma of sample 156.

**Figure 18 foods-11-02553-f018:**
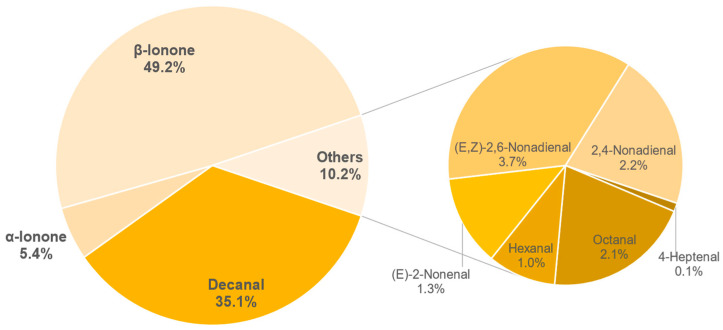
Percentage of OAV of each compound in the total aroma of sample 930.

**Figure 19 foods-11-02553-f019:**
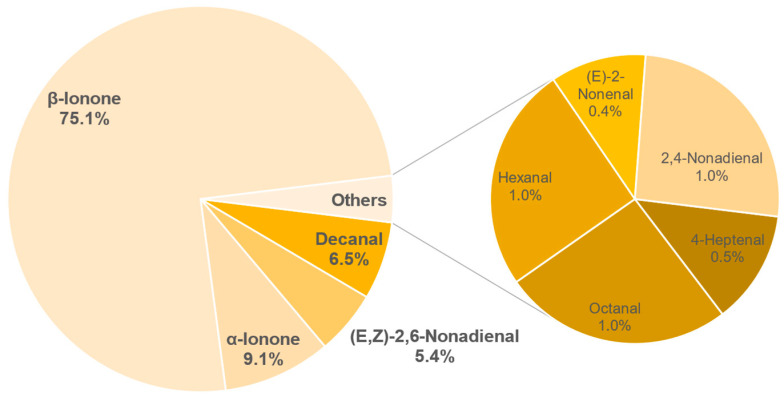
Percentage of OAV of each compound in the total aroma of sample 743.

**Figure 20 foods-11-02553-f020:**
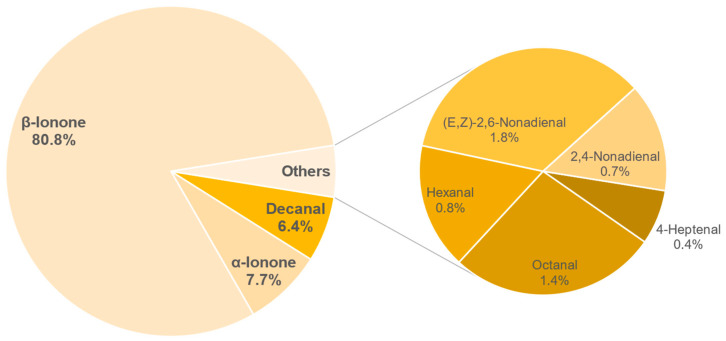
Percentage of OAV of each compound in the total aroma of sample 147.

**Figure 21 foods-11-02553-f021:**
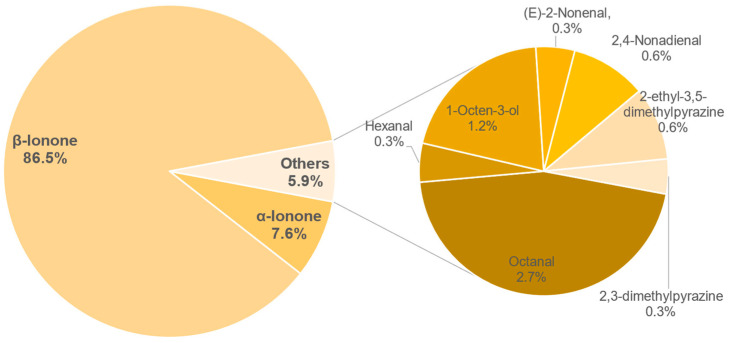
Percentage of OAV of each compound in the total aroma of sample 369.

**Figure 22 foods-11-02553-f022:**
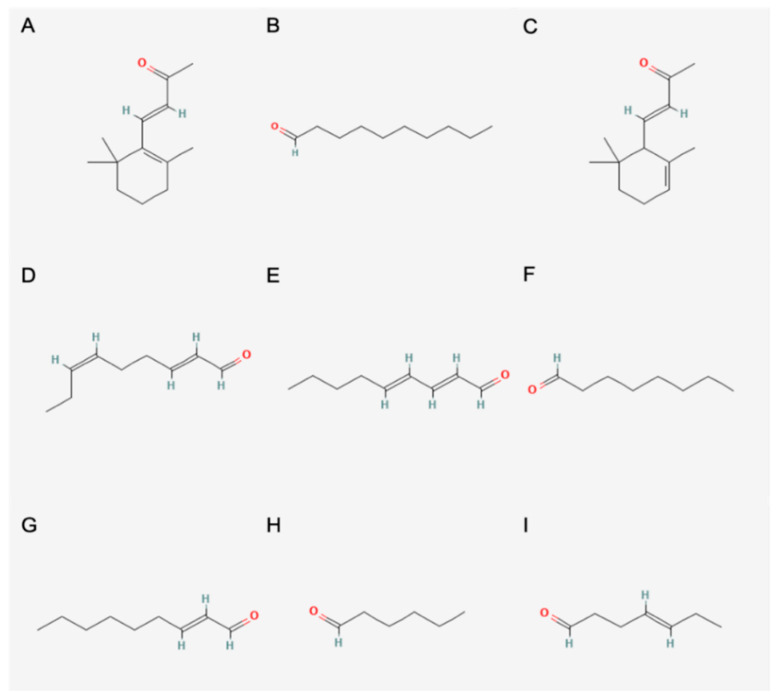
Structure of the main aroma compounds. (**A**): *β*-ionone, (**B**): decanal, (**C**): *α*-ionone, (**D**): (*E*,*Z*)-2,6-nonadienal, (**E**): 2,4-nonadienal, (**F**): octanal, (**G**): (*E*)-2 nonenal, (**H**): hexanal, (**I**): 4-heptenal [56].

**Table 1 foods-11-02553-t001:** Leaf types, their variety, and their collection place.

Leaf Type	*Coffea arabica* Variety	Collection Place
Old leaf	Pacamara	Finca La Palma
Yellow leaf	Pacamara	Finca La Palma
Old leaf	Bourbon Tekisic	Finca La Quintanilla
Young leaf	Bourbon Tekisic	Finca La Quintanilla
Shoot	Bourbon Tekisic	Finca La Quintanilla

**Table 2 foods-11-02553-t002:** Processing steps and their explanation.

Processing	Explanation
None (whole leaf)	No further mechanical intervention.
Blending	The leaves were blended in a kitchen blender. For 100 g leaves, around 400 mL tap water was added.
Cutting	Leaves were cut with a kitchen knife to small strips (20 mm wide).
Rolling	The leaves were rolled by hand.
Freezing	The leaves were frozen in a freezer at −20 °C for 2 days.
Crumbling	Leaves were crumbled by hand.
Steaming	Leaves were steamed in a 50 L pot. A sieve was placed in the centre of the pot and approximately 2 L of tap water was boiled under the leaves. The temperature was measured at the lid of the pot. The process was stopped when the temperature reached around 100 °C.

**Table 3 foods-11-02553-t003:** Moisture content of the different coffee leaves directly after picking shown as mean ± standard deviation.

Leaf Type	Moisture Content [%]
Pacamara yellow	56.28 ± 1.02
Shoots Bourbon whole	72.74 ± 1.73
Bourbon old	62.83 ± 2.22
Bourbon young	72.89 ± 0.99

**Table 4 foods-11-02553-t004:** Produced coffee leaf tea samples in this study including the test number, the variety, the leaf type, the processing, the drying and the fermentation process.

	Test No.	*Coffea arabica* Variety	Leaf Type	Processing	Drying	Fermentation
1	936	Pacamara	old	whole	air	none
2	324	Pacamara	yellow	whole	air	none
3	742	Pacamara	old	crumbled	air	none
4	183	Pacamara	yellow	crumbled	air	none
5	502	Pacamara	old	cutted	air	none
6	643	Pacamara	old	blended	air	none
7	842	Pacamara	old	crumbled	air	Yeast
8	234	Pacamara	old	whole	oven	none
9	238	Pacamara	old	blended	air	*Lactobacillus*
10	182	Pacamara	old	blended	air	Yeast
11	789	Pacamara	old	blended	air	Wild
12	156	Pacamara	yellow	blended	air	Wild
13	687	Pacamara	yellow	blended	air	Yeast
14	463	Pacamara	yellow	blended	air	*Lactobacillus*
15	289	Bourbon	shoot	blended	air	Wild
16	138	Bourbon	old	blended	air	Yeast
17	147	Bourbon	young	blended	air	Wild
18	392	Bourbon	young	steamed/rolled	air	none
19	305	Bourbon	young	rolled/fermented	air	Wild
20	930	Bourbon	shoot	blended/steamed	air	none
21	743	Bourbon	old	whole	air	Wild
22	901	Bourbon	old	whole/frozen	air	none
23	369	Bourbon	old	whole	roasted	none
24	220	Bourbon	shoot	whole	air	none

**Table 5 foods-11-02553-t005:** Results of ranking test and descriptive terms for colour, odour (via orthonasal detection) and flavour (via retronasal detection) of the panel for the best 8 tea samples.

Test Number	Ranking	Colour	Odour	Flavour
687	1	sediment turbid red-brown	peach-like	peach-like
182	2	clear red	floral woody	sweet peach-like
369	3	clear yellow-brown	popcorn-like smoky roasty	popcorn-like roasty
147	4	very clear yellow-brown	chestnut flower-like floral	honey-like grassy
156	5	red clear	honey floral	honey-like acacia flower-like
234	6	clear amber	grassy rooibos-like	green bean-like vegetal
930	7	clear yellow-green	floral sweet	floral basil-like
743	8	turbid light orange	green bean-like	grassy green bean-like broccoli-like

## Data Availability

Data is contained within the article or Appendix A.

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
