# Peer review of "Coffee Leaf Tea from El Salvador: On-Site Production Considering Influences of Processing on Chemical Composition"

_foods, 2022, doi:10.3390/foods11172553_

Round 1
Reviewer 1 Report
The manuscript presents the study of different methods to prepare tea from coffee leaves of two varieties harvested in a community of El Salvador.
This work is of scientific and social relevance since it will allow coffee producers to elaborate a co-product and obtain economic benefits.
The paper is well written and clear, however, it can be improved and the following changes are suggested:
In the text in general revise some details such as the use of commas instead of decimal points in the data with fractions.
Line 183: Explain in more detail the experience of the 7 people who carried out the sensory evaluation. Is it a trained panel? Is your training in coffee or tea?
line 236: Include reference for the calculation or determination of the OAV.
Table 4 to 11: In all tables the error bars are included but the ANOVA result is missing.
Line 268: How was the chlorogenic acid content determined, were standards of the most abundant isomers of the Coffea genus used or was a mixture of chlorogenic acids used? Explain why there was such a large variability in the results of chlorogenic acid content in the different processes.
Author Response
The manuscript presents the study of different methods to prepare tea from coffee leaves of two varieties harvested in a community of El Salvador.
This work is of scientific and social relevance since it will allow coffee producers to elaborate a co-product and obtain economic benefits.
REPLY: Thank you for the assessment of our paper.
The paper is well written and clear, however, it can be improved and the following changes are suggested:
In the text in general revise some details such as the use of commas instead of decimal points in the data with fractions.
REPLY: We have carefully looked throughout the draft and corrected all decimal point issues. Thank you for spotting this mistake.
Line 183: Explain in more detail the experience of the 7 people who carried out the sensory evaluation. Is it a trained panel? Is your training in coffee or tea?
REPLY: The CVUA Karlsruhe is accredited for sensory analysis of both coffee and tea samples. The affiliated institute in Hohenheim has its major expertise in Camellia sinensis tea. We have expanded the information about the panel in the methods section as requested.
line 236: Include reference for the calculation or determination of the OAV.
REPLY: Reference added as requested.
Table 4 to 11: In all tables the error bars are included but the ANOVA result is missing.
REPLY: The statistically significant differences are pointed out in the text.
Line 268: How was the chlorogenic acid content determined, were standards of the most abundant isomers of the Coffea genus used or was a mixture of chlorogenic acids used? Explain why there was such a large variability in the results of chlorogenic acid content in the different processes.
REPLY: As detailed in section 2.2.3, chlorogenic acid is determined by using NMR analysis. We did not include isomers of chlorogenic acid (information about the exact chemical name of chlorogenic acid was added). Chlorogenic acid is discussed in the section 3.6.1. We have included some points about leaf maturity which may explain the wide variations, and this is corroborated by the limited literature available. Otherwise processing may lead to differences. E.g. from normal coffee roasting, it is known that the acids may degrade. This could happen during thermal tea processing as well.
Reviewer 2 Report
Dear Editors,
The article entitled “Coffee Leaf Tea from El Salvador: On-site Production considering Influences of Processing on Chemical Composition” is interesting compiling a lot of data from the laboratory experiment. However, the in general the article is quite hard to follow since the experimental design is not really clear. Some points are need clarifications:
1. Introduction:
a. Please make sure that references No. 11-13 are describing about “coffee leaf tea”, not real tea leaf
b. The purpose of the study is not clearly stated
c. The story of coffee leaf tea in El Salvador has not described
2. Material and Method
a. Table 1: Finca?
b. Table 1: Why the authors did use the sample of Shooter from Pacamara variety and La Palma estate? To have a good experimental design, the samples should be balance.
c. Figure 2 (a-d) must be in a page (do not separate it into two pages)
d. From Fig. 2, it seems that the process was not well-controlled in terms of light, oxygen intervention, cleanness, etc.
e. Table 2: Whole leaf is not process
f. Sensory evaluation: it is not clear why the experimental design is like that. A schematic diagram for explaining the sensory evaluation
3. Results and Discussion
a. There are some results in which the methods had not been explained, e.g., Water content, essential oil content and ash content
b. Figure 4 on-wards it is really confusing which samples that you analyzed? The information of the samples is not clear. Why didn’t you analyse all the samples? I think the discussion is also not too deep.
4. Additional note:
Since this research used human for doing sensory analysis, the ethical clearance document is required.
I suggest a major revision for this article as this article is too complex and do not have a good flow. This might be clearer if the authors separate the article into several articles which is more focus and well-structured.
Author Response
Dear Editors,
The article entitled “Coffee Leaf Tea from El Salvador: On-site Production considering Influences of Processing on Chemical Composition” is interesting compiling a lot of data from the laboratory experiment. However, the in general the article is quite hard to follow since the experimental design is not really clear. Some points are need clarifications:
- Introduction:
- Please make sure that references No. 11-13 are describing about “coffee leaf tea”, not real tea leaf
REPLY: We have clarified the point about tea in the text.
- The purpose of the study is not clearly stated
REPLY: We believe that the last paragraph of the introduction was quite explicit regarding the aims of our study: “In this study, the possibilities to produce coffee leaf tea in a country of origin were investigated. With the locally available resources, different coffee tea samples were produced. Furthermore, the effects of leaf types, coffee varieties, processing, and drying methods on the consumer acceptance, the aroma profile and the product characteristics should be understood and described were studied.”
- The story of coffee leaf tea in El Salvador has not described
REPLY: Coffee production has a long tradition in El Salvador. Since 1880, coffee has been the country's largest export and coffee production is currently the main source of jobs, tax revenue, and environmental activity. At times, El Salvador was considered the most efficient coffee producer in the world, and its coffee production efficiency could even surpass that of the big producing countries, Brazil and Colombia (Paige, 1993). Inv2019, approximately 45600 t of coffee were produced on a cultivated area of 130600vhectares which is 6.3% of the total area of El Salvador (brand eins, 2020). Nevertheless, no study is known about the traditional production of tea from coffee leaves in El Salvador. We believe that we are the first group to produce this tea in El Salvador. So there is not story or history about that product to describe.
References:
Paige, J. M. (1993). Coffee and Power in El Salvador. Latin American Research Review, 28 (3), 7-40
brand eins, s. c. (2020). Kaffee in Zahlen.
- Material and Method
- Table 1: Finca?
REPLY: “Finca” was changed to “Collection place”. The coffee producer La Buena Esperanza was founded 1924 by Amalia Quintanilla and is nowadays managed by Andres Quintanilla in the 4th generation. The two associated plantations (fincas) cover a size of 96.5 hectares and employ 8 people together with 80 harvesters. The Arabica varieties Bourbon Tekisic, Typica, Pacamara, Centroamerica, Icatu and Marsellesa are cultivated with a total annual production of 50 -100 tons per year (Coffee Store, 2021).
Reference:
Coffee Store. (2021). Plantagen URL: https://www.coffee-store.de/UEber-uns/Plantagen/.
- Table 1: Why the authors did use the sample of Shooter from Pacamara variety and La Palma estate? To have a good experimental design, the samples should be balance.
REPLY: Yes, this would have been appreciated from a mathematical standpoint to have a full factorial design. However, the shooters at Finca La Palma had already been cut prior to the arrival of the research team and were unavailable during the study period in January/February 2021.
- Figure 2 (a-d) must be in a page (do not separate it into two pages)
REPLY: This will be done editorially by the copy-editor. Currently, with the necessicity of track changes moving around the layout, we must live with that page break.
- From Fig. 2, it seems that the process was not well-controlled in terms of light, oxygen intervention, cleanness, etc.
REPLY: If you are doing field research in an agricultural setting of a coffee plantation, you have to live with the materials that are available. Probably, a laboratory setting might have been better controllable. However, on the other hand, we want to help coffee farmers to produce these products using the materials they have already available and are economic. Therefore, it would be out of the question to have thermostatted drying devices or other well-controllable equipment. Please think about the fact that many coffee farms probably do not have access to the level of electricity required to power such equipment anyway.
- Table 2: Whole leaf is not process
REPLY: “Whole leaf” was changed to “none” .
- Sensory evaluation: it is not clear why the experimental design is like that. A schematic diagram for explaining the sensory evaluation
REPLY: The authors are at a loss what diagram might be appropriate here. This was a simple process. The samples were tasted by the panel, and then a ranking of 8 samples was conducted.
- Results and Discussion
- There are some results in which the methods had not been explained, e.g., Water content, essential oil content and ash content
REPLY: Thank you for spotting this omission. A new section 2.2.4 was added about NIR measurement of these parameters.
- Figure 4 on-wards it is really confusing which samples that you analyzed? The information of the samples is not clear. Why didn’t you analyse all the samples? I think the discussion is also not too deep.
REPLY: Actually, we did measure all samples. See appendix. The figures summarize the different treatments.
- Additional note:
Since this research used human for doing sensory analysis, the ethical clearance document is required.
RESPONSE: See new inclusion in the section “Institutional Review Board Statement”. According to the legislation in the EU and Germany, the affiliated institutes do not require ethical clearance for sensory analysis of foods. Nevertheless, we considered the IFST guidelines for ethical and professional practices for the sensory analysis of foods [1]. According to that, potential adverse effects for the assessors were excluded as only small amounts of tea were taken into the mouth (about one tasting spoon). The amount of relevant compounds (caffeine, EGCG, chlorigenic acids, trigonelline etc.) in this portion was considerably below toxic levels, even assuming that the assessor would ingest the full portion and not spit it out. All assessors had training in sensory analysis of foods, and regularly performed sensory testing of teas in line of their normal duties. Furthermore, the CVUA Karlsruhe is externally accredited by the national accreditation body of the Federal Republic of Germany (Deutsche Akkreditierungsstelle, DAkkS) for sensoric analysis of foods. The CVUA Karlsruhe is also permanently permitted by German federal state law to conduct sensory testing of foods including teas in its capacity as governmental control laboratory (see [3] and details in Lachenmeier & Monakhova [4]).
References:
[1] https://www.ifst.org/membership/networks-and-communities/special-interest-groups/sensory-science-group/ifst-guidelines
[2] https://www.dakks.de/de/akkreditierte-stelle.html?id=D-PL-18866-02-00
[3] Ministerium Ländlicher Raum: Verwaltungsvorschrift des Ministeriums Ländlicher Raum über die Dienstaufgaben und Zuständigkeitsbereiche der Chemischen und Veterinäruntersuchungsämter und des Staatlichen Tierärztlichen Untersuchungsamtes Aulendorf – Diagnostikzentrum [Administrative regulation of the Ministry of Rural Affairs regarding the official duties and jurisdiction of the Chemical and Veterinary Investigation Laboratories and the State Veterinary Laboratory Aulendorf - center of diagnostic investigations]. GABl 2000, 2000:358-359.
[4] Lachenmeier, D.W., Monakhova, Y.B. Short-term salivary acetaldehyde increase due to direct exposure to alcoholic beverages as an additional cancer risk factor beyond ethanol metabolism. J Exp Clin Cancer Res 30, 3 (2011). https://doi.org/10.1186/1756-9966-30-3
I suggest a major revision for this article as this article is too complex and do not have a good flow. This might be clearer if the authors separate the article into several articles which is more focus and well-structured.
REPLY: The authors do not like slicing research into several articles, purely to increase citations indices. We believe that our full presentation of all analyses conducted on these samples is justified, and the open access journal FOODS also does not have word limits as might be the case for classical print journals. We have moved some information into the annex as suggested by the other reviewers, so that the main part is shorter and the flow hopefully more clear. Apart from that, we believe that the structure of our article is quite clear and has the same flow in all parts of the manuscript:
Reviewer 3 Report
The current manuscript presented by Steger et al. is aimed to assess various factors affecting the phytochemical composition of coffee by-products, especially leaves. It highlights the requirements of the EU for novel and non-traditional foods. Despite the research addressed many points, it lacks various aspects to be a scientific paper, for examples:
- The manuscript seems to be a report more than a paper,
- The abstract needs a revision, research significance, and a conclusion, as future outlook as well,
- The introduction is quite general,
- The material and Methods has a lot of defects such as no specifications were mentioed for the used devices. Also, it needs extensive organizational,
- All plants pictures must be moved as supplementary materials,
- The results needs to be clearly organised and discussed. Also, they are just reporting the obtained results. More statistical calculations, PLS modeling, Factorial design experiments,... are required.
- Some figures have bad resolution,
Finally, the authors should specify some significant points which are highlighted within the text, and the others are moved to Suppl. Materials.
Appendix part: No comments!
Author Response
The current manuscript presented by Steger et al. is aimed to assess various factors affecting the phytochemical composition of coffee by-products, especially leaves. It highlights the requirements of the EU for novel and non-traditional foods. Despite the research addressed many points, it lacks various aspects to be a scientific paper, for examples:
- The manuscript seems to be a report more than a paper,
REPLY: What is a paper? We believe that our article has all elements justified to be published as MDPI article type “article”.
- The abstract needs a revision, research significance, and a conclusion, as future outlook as well,
REPLY: A concluding remark and outlook was added to the abstract as requested. We believe that the research aim and results are adequately covered already.
- The introduction is quite general,
REPLY: According to the authors guideline: “The introduction should briefly place the study in a broad context and highlight why it is important.“. The authors believe to have achieved this aim.
- The material and Methods has a lot of defects such as no specifications were mentioed for the used devices. Also, it needs extensive organizational,
REPLY: This is no analytical chemistry paper, but an applied food science paper. All used methods are either established standard norm procedures or have been published separately by the authors including method validation data. Therefore, we believe that in this applied paper, which is already quite long, a repetition of methodology is unnecessary. You also always have the problems of self-plagiarism and it makes not much sense to rephrase your own text, which is rather redundant anyway.
- All plants pictures must be moved as supplementary materials,
REPLY: Why? We believe that an adequate illustration is necessary and appropriate. See, for example, our previous paper on the topic: Blumenthal P, Steger MC, Quintanilla Bellucci A, Segatz V, Rieke-Zapp J, Sommerfeld K, Schwarz S, Einfalt D, Lachenmeier DW. Production of Coffee Cherry Spirits from Coffea arabica Varieties. Foods. 2022; 11(12):1672. https://doi.org/10.3390/foods11121672
- The results needs to be clearly organised and discussed. Also, they are just reporting the obtained results. More statistical calculations, PLS modeling, Factorial design experiments,... are required.
REPLY: Yes, the results section is reporting the results. We also discuss the results in light of the previous, if rather limited, literature. As the text is already rather long, we feel hesitant to add unnecessary detail on standard methodologies.
- Some figures have bad resolution,
REPLY: In our original word file, all figures are vector files in high quality. Probably conversion to PDF changed the resolution. This should be dealt with during copy-editing.
Finally, the authors should specify some significant points which are highlighted within the text, and the others are moved to Suppl. Materials.
REPLY: We are at a loss here, which material should be moved around. Hopefully, the changes requested by the other reviewers, due to which some material moved into the annex, is adequate. We also do not believe that the MDPI journal layout allows highlighting of text.
Appendix part: No comments!
Reviewer 4 Report
Dear authors,
There is no doubt about the work behind the manuscript. I believe it is an interesting research work that must be shown to the scientific community. However, under my opinion, several main aspects must be clarified. First, I am not able to clearly define in one sentence the final objective of the research beyond to increase knowledge about such specific material. Moreover, if the interest is to provide information about the chemical composition of the product available to the consumer, why the analysis of the phenolic compounds is performed using raw material and not the beverage? Chemical changes may occur during the processes to obtain coffee leaf tea.
Beside such concerns and, with the main purpose to help you increasing the quality of your manuscript, I would like you consider the following suggestions:
Abstract must be review focusing on the caffeine content (lines 26 and 27). Moreover, I would appreciate some conclusion following the proposed objectives directed to the specific interest on such food product.
Introduction
Line 42: Why is there an unstable price condition? I would emphasize in the relevance of by-product valorisation.
Line 44: What do you mean with “sustainable future”? in which terms?
Line 64- The sentence makes to understand the reader that the beneficial bioactive properties were observed in the references 14-17 however, it seems that just reference 16 is focused on such topic. Please, review the information accuracy. I suggest to firstly write the compounds found in the leaves and later justify the observed their beneficial properties.
Line 77- As a suggestion, would it be interesting to highlight that even the commercialization of coffee leaf tea was approved by EU Commission there is a limitation regarding the content of caffeine, chlorogenic acid and epigallocatechin gallate in the matrix or beverage?
Line 79- I would specify the purpose of such investigation.
Material and methods
-As a curiosity: How many plants were sampled?
-I would include all the images in the supplementary material. Nevertheless, tables must be placed before figures since it is the mentioned order in the text.
-It would be helpful for the reader to show a flow diagram of the leaves processing.
-Is the oven drying method including a circulating air?
-2.1.2 – I suggest rewriting this section for a better understanding of the two fermentation processes carried on (flow diagram would extremely help to understand the treatments).
-2.1.3- which are those samples that were not mixed before?
In general terms, the description of the treatments/processing (mostly the steps order) is not clear under my opinion.
-2.2.1 – I would place this section at the end of 2.1.
Line 155-160- It is not necessary to explain how to filter a sample for HPLC analysis. Some relevant information is missing.
Line 162-169- There is some relevant information missing, i.e., the standard or the wavelength used.
2.3. There is a controversy about how many panelists are needed to obtain consistent and stable results for descriptive tests however, several publications pointed that at least 8 panelists are needed (preferably 10). Moreover, in preference test it is recommended around 20-30 panelists so your test results carry statistical significance. Would it be possible to increase the number of panelists in your study?
2.4.4. can be included in 2.4.3.
2.4.5. it is also considered sensory analysis. I suggest restructuring the section material and methods.
Results
In general terms, results section must be improved regarding the structure (some subsections must be placed on M&M) and with the purpose to clarify the achievements. Moreover, figures must be also improved following the formatting of a scientific article. However, one of the most important aspects is that results about ash, essential oil, lactic acid, acetic acid (and other) content is given but no explanation about how such determinations were performed is included in M&M. Moreover, it could be interesting to reinforce the connection between the chemical characterization and sensory analysis.
Summarizing, the manuscript must be improved to stablish a clear objective to achieve, including all the information to make reproducible your research, stablishing a coherent structure and clarifying a main conclusion (beside the formatting aspects).
Best Regards
Author Response
Dear authors,
There is no doubt about the work behind the manuscript. I believe it is an interesting research work that must be shown to the scientific community. However, under my opinion, several main aspects must be clarified. First, I am not able to clearly define in one sentence the final objective of the research beyond to increase knowledge about such specific material. Moreover, if the interest is to provide information about the chemical composition of the product available to the consumer, why the analysis of the phenolic compounds is performed using raw material and not the beverage? Chemical changes may occur during the processes to obtain coffee leaf tea.
REPLY: Thank you for the assessment of our paper. The aim was to investigate if it is at all possible to conduct coffee leaf tea production in the setting of a coffee plantation in El Salvador. It is true that chemical parameters in tea (as well as in coffee) are always determined in respect to the dry material. This is a convention found in almost all ISO, EN and DIN norms regarding coffee or tea analysis. Probably, by analyzing the dry material the results are much more stabile and comparable than when analyzing the beverage. This probably stems because of the manyfold preparation techniques, extraction influences, temperatures, water quality, brewing ratios etc. For example, compare an espresso or a filter coffee. For tea, there is also no international convention about preparing a beverage. And we clearly have no standard for coffee leave tea, which is a novel product only emerging on the market.
Beside such concerns and, with the main purpose to help you increasing the quality of your manuscript, I would like you consider the following suggestions:
Abstract must be review focusing on the caffeine content (lines 26 and 27). Moreover, I would appreciate some conclusion following the proposed objectives directed to the specific interest on such food product.
REPLY: We have changed the presentation of caffeine content. A last sentence regarding outlook and interest of the product was added.
Introduction
Line 42: Why is there an unstable price condition? I would emphasize in the relevance of by-product valorisation.
REPLY: It is true that coffee prices on the world markets are currently high, but the profits are not made by the farmers. Therefore, it is certainly true that coffee farmers are experiencing economic pressures. We have revised the sentence to better reflect this argument.
Line 44: What do you mean with “sustainable future”? in which terms?
REPLY: The sentence was revised.
Line 64- The sentence makes to understand the reader that the beneficial bioactive properties were observed in the references 14-17 however, it seems that just reference 16 is focused on such topic. Please, review the information accuracy. I suggest to firstly write the compounds found in the leaves and later justify the observed their beneficial properties.
REPLY: We revised these sentences by toning down the health effects, which are mostly not based on clinical studies, but are mostly traditional observations or in vitro studies. We also changed and condensed the section about compounds in the leaves.
Line 77- As a suggestion, would it be interesting to highlight that even the commercialization of coffee leaf tea was approved by EU Commission there is a limitation regarding the content of caffeine, chlorogenic acid and epigallocatechin gallate in the matrix or beverage?
RESPONSE: The limits were added to the introduction as requested.
Line 79- I would specify the purpose of such investigation.
RESPONSE: The aim was expanded as requested.
Material and methods
-As a curiosity: How many plants were sampled?
REPLY: About 1000 per variety. Information was added to the materials section.
-I would include all the images in the supplementary material. Nevertheless, tables must be placed before figures since it is the mentioned order in the text.
REPLY: We would like to retain the images in the main text. The sequence of tables and figures was reordered according to the one in the text.
-It would be helpful for the reader to show a flow diagram of the leaves processing.
REPLY: Basically, there is only one step for each preparation. The processing is already detailed in Table 2.
-Is the oven drying method including a circulating air?
RESPONSE: Yes. Information was added to the materials section.
-2.1.2 – I suggest rewriting this section for a better understanding of the two fermentation processes carried on (flow diagram would extremely help to understand the treatments).
RESPONSE: We are at a loss how to improve the text. The fermentation was an extremely simple process. The microorganisms were mixed in water. The mixture was sprayed on the leaves. The leaves were then placed in closed buckets. We have reordered the text into the actual sequence for better understanding.
-2.1.3- which are those samples that were not mixed before?
RESPONSE: The sample treatment is stated in table 2. Some of the samples were already blended during processing. We have clarified the text with a reference to table 2.
In general terms, the description of the treatments/processing (mostly the steps order) is not clear under my opinion.
RESPONSE: We hope that the additions now make the steps clear. Mostly they were very simple processes as stated in table 2.
-2.2.1 – I would place this section at the end of 2.1.
RESPONSE: No, we disagree here. The steps in section 2.1 were done in El Salvador. The steps in section 2.2.1 were done in the laboratory in Germany. We believe that these two processes should stay separated.
Line 155-160- It is not necessary to explain how to filter a sample for HPLC analysis. Some relevant information is missing.
RESPONE: The reviewer is correct. We deleted unnecessary methodological details.
Line 162-169- There is some relevant information missing, i.e., the standard or the wavelength used.
REPONSE: This is the standard Folin-Ciocalteu procedure. Similarly to above, we have condensed unnecessary details regarding standard methods.
2.3. There is a controversy about how many panelists are needed to obtain consistent and stable results for descriptive tests however, several publications pointed that at least 8 panelists are needed (preferably 10). Moreover, in preference test it is recommended around 20-30 panelists so your test results carry statistical significance. Would it be possible to increase the number of panelists in your study?
RESPONSE: The reviewer is correct that the more the better in sensory analysis. Due to COVID-19 pandemic contact restrictions at the time of the sensory analysis of the samples (lockdown in Germany), we just could not increase the number of tasters. However, consensus could be reached in all cases and there were not large discrepancies that would have made further testing necessary.
2.4.4. can be included in 2.4.3.
RESPONSE: The sections were merged as requested.
2.4.5. it is also considered sensory analysis. I suggest restructuring the section material and methods.
RESONSE: We believe that GC-Olfactometry is something different than classical sensory analysis. Hence, we believe the parts should stay separated.
Results
In general terms, results section must be improved regarding the structure (some subsections must be placed on M&M) and with the purpose to clarify the achievements. Moreover, figures must be also improved following the formatting of a scientific article.
RESPONSE: We carefully looked through the results section, but were unable to identify material that could be moved into the materials section. The figures look fine in the original word file
However, one of the most important aspects is that results about ash, essential oil, lactic acid, acetic acid (and other) content is given but no explanation about how such determinations were performed is included in M&M.
RESPONSE: The determination was done using NIR. The information was introduced into the methods section, see above.
Moreover, it could be interesting to reinforce the connection between the chemical characterization and sensory analysis.
RESPONSE: Yes. This would be interesting for further research. We have currently not approached this issue, such as using multivariate analysis, also considering the restrictions of the sensory analysis discussed above. We have not conducted a fully quantitative analysis, which would allow for finding correlations.
Summarizing, the manuscript must be improved to stablish a clear objective to achieve, including all the information to make reproducible your research, stablishing a coherent structure and clarifying a main conclusion (beside the formatting aspects).
REPONSE: We thank the reviewer for the detailed remarks and improvements of our paper.
Best Regards
Round 2
Reviewer 2 Report
The authors has responded the comments very well.
Author Response
RESPONSE: Thank you for re-assessing our revisions.
Reviewer 3 Report
I think the previous comments did not convince the authors, and therefore, the manuscript is still unsuitable.
Author Response
RESPONSE: Thank you for the second review round of our paper. In light of the comments of the editor and reviewer #3, we have carefully revised our paper and hope that the manuscript is now suitable for publication.
Reviewer 4 Report
Dear authors,
I appreciate you answered all my comments, with which I agree for the most part. However, I must insist in several aspects:
- Confusion about the leaf’s treatments.
Are the procedures described in Table 2 those related to "pre-treatment"? Drying methods (section 2.1.1) are procedures applied to those pre-treated leaves being air-drying the methodology to dry non-treated leaves?
- Regarding my previous comment “2.2.1 – I would place this section at the end of 2.1.”
RESPONSE: No, we disagree here. The steps in section 2.1 were done in El Salvador. The steps in section 2.2.1 were done in the laboratory in Germany. We believe that these two processes should stay separated.
Although, I would prefer to move such information, it is also fine to maintain it there in section 2.2.1. However, it must be irrelevant where the experimental work is done in research collaborations unless the information is shorted following such criteria (which it is not the case because it was not mentioned at any previous point) and it constitutes a relevant aspect to consider.
-Regarding my previous comment “2.3. There is a controversy about how many panelists are needed to obtain consistent and stable results for descriptive tests however, several publications pointed that at least 8 panelists are needed (preferably 10). Moreover, in preference test it is recommended around 20-30 panelists so your test results carry statistical significance. Would it be possible to increase the number of panelists in your study?”
RESPONSE: The reviewer is correct that the more the better in sensory analysis. Due to COVID-19 pandemic contact restrictions at the time of the sensory analysis of the samples (lockdown in Germany), we just could not increase the number of tasters. However, consensus could be reached in all cases and there were not large discrepancies that would have made further testing necessary.
I completely understand the COVID struggle however, the minimum number of panelists is a consensus to statistically validate the obtained results. If increase the “n” is not possible, I strongly suggest the authors to include a reference supporting the “n” selected for this work, to make a note showing such limitation or to talk about tentative results.
In addition, I would like you consider the following comments:
-Statistical significance labels must be indicated in the figures (which they are still blurred, check the resolution please). Moreover, it must be clearly written in the text avoiding expressions such as “statistical indication”
-Some error bars are missing in the figures (in example figure 4) however, following the text, the statistical analysis was performed (so, replicates must exist).
-Please, review the axes titles of figure 4b. Is there two different units? If so, two Y-axis must be included.
-There is no need to write, as an example, “Caffeine [g/(100g) DM]”, it is correct to write “Caffeine (g/100g DM)” (applied to all figures)
-Please, normalize the number of decimals along the text.
-I suggest not to start a section with a figure without any introduction (it is section 3.6.1. , 3.6.2., and so on). It can be interesting to place together the figures related to organic acids.
- Standard deviations are notably high in some cases, please check for outliers’ values.
-Some grammatical/typing mistakes were noticed, please check carefully he manuscript for their correction.
Best Regards,
Author Response
Dear authors,
I appreciate you answered all my comments, with which I agree for the most part. However, I must insist in several aspects:
- Confusion about the leaf’s treatments.
Are the procedures described in Table 2 those related to "pre-treatment"? Drying methods (section 2.1.1) are procedures applied to those pre-treated leaves being air-drying the methodology to dry non-treated leaves?
REPONSE: We have changed the term “pre-treatment” with “processing” throughout to clarify that these procedures were the processing of the leaves on the Coffee plantation in El Salvador.
- Regarding my previous comment “2.2.1 – I would place this section at the end of 2.1.”
PREVIOUS RESPONSE: No, we disagree here. The steps in section 2.1 were done in El Salvador. The steps in section 2.2.1 were done in the laboratory in Germany. We believe that these two processes should stay separated.
Although, I would prefer to move such information, it is also fine to maintain it there in section 2.2.1. However, it must be irrelevant where the experimental work is done in research collaborations unless the information is shorted following such criteria (which it is not the case because it was not mentioned at any previous point) and it constitutes a relevant aspect to consider.
RESPONSE: Yes, this is true. Basically, it does not matter where the analysis is conducted. We only want to be clear what is processing on the coffee farm, and what is sample preparation in the lab (wherever located). By eliminating the term “pre-treamtent”, which is also often applied in analytical chemistry for “sample pre-treatment”, we now hope that the difference is clear.
-Regarding my previous comment “2.3. There is a controversy about how many panelists are needed to obtain consistent and stable results for descriptive tests however, several publications pointed that at least 8 panelists are needed (preferably 10). Moreover, in preference test it is recommended around 20-30 panelists so your test results carry statistical significance. Would it be possible to increase the number of panelists in your study?”
PREVIOUS RESPONSE: The reviewer is correct that the more the better in sensory analysis. Due to COVID-19 pandemic contact restrictions at the time of the sensory analysis of the samples (lockdown in Germany), we just could not increase the number of tasters. However, consensus could be reached in all cases and there were not large discrepancies that would have made further testing necessary.
I completely understand the COVID struggle however, the minimum number of panelists is a consensus to statistically validate the obtained results. If increase the “n” is not possible, I strongly suggest the authors to include a reference supporting the “n” selected for this work, to make a note showing such limitation or to talk about tentative results.
RESPONSE: The limitations in the taste panel were now noted in the methods and results sections, and the results were marked as preliminary as requested.
In addition, I would like you consider the following comments:
-Statistical significance labels must be indicated in the figures (which they are still blurred, check the resolution please). Moreover, it must be clearly written in the text avoiding expressions such as “statistical indication”
RESPONSE: All claims about statistical signifance in the text are backed up by ANOVA analysis. See methods section: The calculations (One way ANOVA (confidence level p < 0.05) were applied using SPSS (IBM Corporation, Armonk, USA). Otherwise, the figures look fine in our word version at 500% magnification. We hope that the copy editor can tackle this out. With a deadline of 3 days for the revision, we are currently not able to re-touch the figures otherwise.
-Some error bars are missing in the figures (in example figure 4) however, following the text, the statistical analysis was performed (so, replicates must exist).
RESPONSE: The standard deviations are given in the data annexes at the end of the paper. The standard deviations, specifically of NMR analysis are so low that it does not make sense to introduce error bars that are not visible. Additionally, we would need more time to re-touch the figures.
In the instances of figure 4, where no error bars are shown, only one sampe had been prepared using this processing. The error bars in our figures do not shown the methodological error of replicate analysis of the same sample, but the differences between actual replicates on the farm using these methods. This alone explains the rather large error bars in some instances.
-Please, review the axes titles of figure 4b. Is there two different units? If so, two Y-axis must be included.
RESPONSE: The units are correct, as there are different units for water and essential oil. We have clarified in the legend of figure 4.
-There is no need to write, as an example, “Caffeine [g/(100g) DM]”, it is correct to write “Caffeine (g/100g DM)” (applied to all figures)
RESPONSE: We did not manage to re-touch the figures within the deadline of the editorial office. Perhaps this can be tackled out by the copy editor if absolutely necessary.
-Please, normalize the number of decimals along the text.
RESPONSE: We not necessarily agree with this request. All methods have different sensitivities, so that a different number of significant decimals could result between the different methods. E.g. protein could be stated as 96.1%, because the second decimal is not significant in the method, but some minor flavour compound could still be reported as 0.001%. Anyway, we carefully checked the decimal usage and corrected some minor mistakes.
-I suggest not to start a section with a figure without any introduction (it is section 3.6.1. , 3.6.2., and so on). It can be interesting to place together the figures related to organic acids.
RESPONSE: We have carefully moved the text so that each section starts with text rather than with figure or table. It is, however, impossible to show the organic acids on one page, so we believe the current flow is better.
- Standard deviations are notably high in some cases, please check for outliers’ values.
RESPONSE: It must be noted that the samples were prepared under the time-pressure of the short period of the coffee picking campaign on a real coffee farms. For that, we believe that the standard deviations are not unusual for natural variations, e.g. between varieties. We also re-checked the analytical replicates, and also did not detect anything unusual.
-Some grammatical/typing mistakes were noticed, please check carefully he manuscript for their correction.
RESPONSE: We have carefully re-checked the spelling of the article. We also corrected some mistakes in the reference list.